# The Source Structure Design of the Rotating Magnetic Beacon Based on Phase-Shift Direction Finding System

**DOI:** 10.3390/s22218304

**Published:** 2022-10-29

**Authors:** Bo Li, Binfeng Yang, Fenghua Xiang, Jiaojiao Guo, Hailin Li

**Affiliations:** Information and Navigation College, Air Force Engineer University, Xi’an 710077, China

**Keywords:** phase-shift angle measurement, rotating magnetic, magnetic beacon, COMSOL simulation, optimization design

## Abstract

Target azimuth information can help further improve the accuracy of magnetic orientation, but the current periodic magnetic field generated by the magnetic beacon is multivalued, so it is not suitable for azimuth measurement. According to the distribution of a rotating magnetic field and the phase angle measuring principle, we put forward a new magnetic source structure design of a multiple rotating permanent magnet array by adjusting the spacing d, the rotating speed ω and the initial rotation angle φ, and then verified the mathematical model using COMSOL simulation software. A triple structure was obtained by comparison ($d_{3}=\sqrt{3}d_{1}=\sqrt{3}d_{2}=4\sqrt{3}\;m$, $d_{3}=\sqrt{3}d_{1}=\sqrt{3}d_{2}=4\sqrt{3}\;m$, $\varphi_{1}=0$, $\varphi_{2}=\frac{4\pi }{5}\;rad$. $\varphi_{3}=\pi \;rad$), which can produce a strong characteristic magnetic signal similar to a heart-shaped field pattern. Finally, a signal transceiver system was set up for the experiment. The experimental result shows that the waveform of the magnetic signal generated by the real beacon meets the requirement of having a unique maximum value and good directivity within a period, which proves the practical application effect of the structure.

## 1. Introduction

Compared to other navigation methods, magnetic navigation [1,2,3] has the advantages of autonomy, no accumulated error, and strong anti-interference ability, which reduces the risk of the signal being captured and attacked.

According to different magnetic sources, magnetic navigation can be divided into geomagnetic navigation and artificial magnetic navigation [4]. Geomagnetic navigation relies on geodesy to match geophysical features, including geomagnetic matching [5,6], terrain matching [7], and gravity matching [8,9]. However, the matching technology itself requires the support of a high-precision measuring instrument, a sizable and perfect geomagnetic field model, and a mature positioning algorithm, so its positioning accuracy makes it difficult to satisfy certain situations which have high positioning requirements. With the progress of material technology, new strong magnetic materials [10,11,12] have brought a new direction to magnetic positioning research. It has been found that the low frequency rotating magnetic field [13,14,15,16] obtained by rotating an artificial magnetic beacon has excellent properties such as high penetration, strong robustness, easy extraction, and is suitable for navigation and positioning applications in complex scenes. Paperno [17] proposed a new magnetic position and direction tracking method based on the quasi-static rotating magnetic field, which improved the speed and accuracy of magnetic tracking. The University of Michigan [18] proposed an electromagnetic beacon and inertial navigation sensor positioning technology, which greatly improves the accuracy of autonomous navigation positioning. Domestically, Zhang Dacheng [19] designed the software and hardware for the magnetic beacon positioning system and used the four-parameter sinusoidal signal reconstruction method to improve identification accuracy. With the help of the magnetic field gradient tensor algorithm, Deng Guoqing [20] applied the magnetic beacon positioning to the real-time positioning of horizontal directional drilling, which improved the positioning accuracy. Wang Run [21] designed and optimized the source structure based on the magnetic beacon structure, which provided an effective solution for the limited transmission distance of magnetic navigation and positioning signals and the difficulty of signal extraction. Optimizing the magnetic source structure or improving the positioning algorithm notwithstanding, the target is located by measuring the amplitude of the magnetic field. However, the measurement of the amplitude needs to solve problems such as a chaotic electromagnetic environment, a strong abnormal field, and quick attenuation of magnetic signal strength; the improvement of the positioning accuracy is increasingly difficult.

Differing from the amplitude of the magnetic signal, the phase information of a low-frequency magnetic signal [22,23,24,25,26] has the advantage of being hard to disturb by the environmental magnetic field. In the radio navigation system [27,28], the phase measurement is an important navigation means which has the advantages of convenience, flexibility, and high precision, and can be combined with other positioning methods to effectively improve the positioning accuracy of navigation equipment. Therefore, it is proposed that the phase information of a low-frequency magnetic signal should be used to measure the orientation of the target and to fuse the information, so as to further improve the magnetic positioning accuracy.

The direction-finding method proposed in this paper refers to the phase-shift angle measuring method in radio navigation systems and the special heart-shaped direction pattern generated by the Voltaic beacon antenna. The research of this method can be divided into six sections: (1) Section 1: We introduce the relevant research results and leads to the research content of the paper; (2) Section 2: We analyze the magnetic field distribution of a low-frequency rotating magnetic field [29] and establish the mathematical model of a magnetic field. Then, the design idea of a magnetic beacon is discussed; (3) Section 3: We design and establish the mathematical models of a binary array, ternary array, and quaternary array. Then, by adjusting the rotation parameters of each permanent magnet, the magnetic field distribution of each array structure under different conditions is compared and analyzed. Finally, COMSOL Multiphysics software is used to verify the optimal structural parameters of each array; (4) Section 4: We simulate the optimized structure in different transmission media using the COMSOL Multiphysics software and verify whether the structure can play a role in different environments; (5) Section 5: We carry out the confirmatory experiment with the help of the structural entity; (6) Section 6: We summarize and evaluate all the work, which provides a better solution for the source design of the direction-finding system based on the rotating magnetic beacon and further advances the research on the direction-finding mechanism.

## 2. Phase-Based Goniometric Principle

### 2.1. Principle of Signal Generation of Rotating Permanent Magnet Beacon

Let m denote the magnetic moment, then the magnetic moment determinant of the permanent magnet is expressed as:(1)$$m=(B_{r}/\mu )V$$

B_r_ is the remanent magnetic intensity of the permanent magnet, V is the volume of the permanent magnet, and μ is the spatial permeability. The static magnetic field produced by a permanent magnet is determined by its magnetic moment, independent of its shape. The static magnetic field generated by a static magnetic dipole is only related to its magnetic dipole distance, which is expressed as l, and the relationship between the magnetic dipole distance and magnetic moment is as follows:(2)$$q_{m}l=\mu m$$
where $q_{m}$ stands for magnetic charge. Equivalent relation can be obtained from Equations (1) and (2):(3)$$q_{m}l=B_{r}V$$

So, a permanent magnet can be equivalent to a magnetic dipole, and in the same way, a rotating permanent magnet can be thought of as a magnetic dipole rotating. A single circular micro electric current is defined as a magnetic dipole, as shown in Figure 1. Supposing I is the current size, R is the radius of the circular electric current, P_m_ is the average molecular magnetic moment and points in the Z-axis direction; according to Maxwell’s equations and Biot–Savart Law, any point Q(r,θ,z) on the circle electric current can produce magnetic induction intensity at the space $P(R,\theta_{1},z)$:(4)$$dB=\frac{\mu }{4\pi }\frac{Idl\times a}{a^{3}},\left|a\right|\neq 0$$

In Equation (4), a represents the distance between $P(R,\theta_{1},z)$ and Q(r,θ,z):$\left|a\right|=\sqrt{r^{2}+R^{2}-2R\times cos(\theta -\theta_{1})+z^{2}}$.

The magnetic field component can be expressed in polar coordinates:(5)$$\left\{\begin{array}{c} B_{x}=\frac{3\mu }{4\pi }\times \frac{P_{m}}{{\left(\sqrt{R^{2}+r^{2}+z^{2}}\right)}^{5}}\times zrcos\theta \\ B_{y}=\frac{3\mu }{4\pi }\times \frac{P_{m}}{{\left(\sqrt{R^{2}+r^{2}+z^{2}}\right)}^{5}}\times zrsin\theta \\ B_{z}=\frac{3\mu }{4\pi }\times \frac{P_{m}}{{\left(\sqrt{R^{2}+r^{2}+z^{2}}\right)}^{5}}\times (2R^{2}+2z^{2}-r^{2}) \end{array}\right.$$

The distance between the general measuring point and the permanent magnet allows R≪r, and then the magnetic field component can be expressed as:(6)$$\left\{\begin{array}{c} B_{x}=\frac{3\mu }{4\pi }\cdot \frac{P_{m}}{{r^{\prime }}^{5}}\cdot zx\\ B_{y}=\frac{3\mu }{4\pi }\cdot \frac{P_{m}}{{r^{\prime }}^{5}}\cdot zy\\ B_{z}=\frac{3\mu }{4\pi }\cdot \frac{P_{m}}{{r^{\prime }}^{5}}\cdot (3z^{2}-{r^{\prime }}^{2}) \end{array}\right.$$
where $r\prime^{2}=x^{2}+y^{2}+z^{2}$.

Since the magnetic moment of a permanent magnet is a vector and changes from moment to moment during rotation, the transformation of the coordinate system should be considered when calculating the magnetic field distribution of a rotating permanent magnet to ensure that the expression of the magnetic field component at any moment can be expressed by a coordinate system, as shown in Figure 2.

Given a permanent magnet in any state, we assume that the initial coordinate system is O-X_1_Y_1_Z_1_, and the magnetic moment of the permanent magnet points in the positive direction Z_1_. Then the initial coordinate system is rotated by an angle φ around axis X_1_, and we can obtain a new coordinate system O-X_2_Y_2_Z_2_, and take this coordinate as the base. The transformation process of the coordinate system can be expressed as follows:(7)$$\left[\begin{array}{c} x_{2}\\ y_{2}\\ z_{2} \end{array}\right]=R_{x}\cdot \left[\begin{array}{c} x_{1}\\ y_{1}\\ z_{1} \end{array}\right]=\left[\begin{array}{ccc} 1 & 0 & 0\\ 0 & cos\varphi & sin\varphi \\ 0 & -sin\varphi & cos\varphi \end{array}\right]\cdot \left[\begin{array}{c} x_{1}\\ y_{1}\\ z_{1} \end{array}\right]$$

Therefore, in the rotation process, coordinate system O-X_1_Y_1_Z_1_ can be expressed by coordinate system O-X_2_Y_2_Z_2_ as:(8)$$\left\{\begin{array}{c} x_{1}=x_{2}\\ y_{1}=y_{2}cos\varphi +z_{2}sin\varphi \\ z_{1}=z_{2}cos\varphi -y_{2}sin\varphi \end{array}\right.$$

Substituting the above equation into the magnetic field intensity distribution Equation (6) yields expression (9) for the magnetic field intensity component in the coordinate system O-X_1_Y_1_Z_1_; the coordinate base of this component is X_2_Y_2_Z_2_, where we let $B^{\prime }=\frac{\mu P_{m}}{4\pi r^{5}}$.
(9)$$\left\{\begin{array}{c} B_{x_{1}}=3B^{\prime }\cdot x_{2}(z_{2}cos\varphi -y_{2}sin\varphi )\\ B_{y_{1}}=3B^{\prime }\cdot (y_{2}cos\varphi +z_{2}sin\varphi )(z_{2}cos\varphi -y_{2}sin\varphi )\\ B_{z_{1}}=B^{\prime }\cdot [2{\left(z_{2}cos\varphi -y_{2}sin\varphi \right)}^{2}-x_{2}{}^{2}-{\left(y_{2}cos\varphi +z_{2}sin\varphi \right)}^{2}] \end{array}\right.$$

On the basis of the projection relationship between coordinate systems, coordinate system O-X_1_Y_1_Z_1_ is projected onto coordinate system O-X_2_Y_2_Z_2_, so the component of the magnetic field intensity in the coordinate system O-X_2_Y_2_Z_2_ can be expressed as:(10)$$\left\{\begin{array}{c} B_{x_{2}}=B_{x_{1}}\\ B_{y_{2}}=B_{y_{1}}cos\varphi -B_{z_{1}}sin\varphi \\ B_{z_{2}}=B_{z_{1}}cos\varphi +B_{y_{1}}sin\varphi \end{array}\right.$$

Substituting this equation back to Equation (9), the magnetic field intensity component measured in the original coordinate system O-X_2_Y_2_Z_2_ can be expressed as:(11)$$\left\{\begin{array}{c} B_{x_{2}}=B^{\prime }\cdot (3x_{2}z_{2}cos\varphi -3x_{2}y_{2}sin\varphi )\\ B_{y_{2}}=B^{\prime }\cdot \left[6y_{2}z_{2}cos\varphi +(x_{2}{}^{2}-2y_{2}{}^{2}+z_{2}{}^{2})sin\varphi \right]\\ B_{z_{2}}=B^{\prime }\cdot \left[(2z_{2}{}^{2}-y_{2}{}^{2}-x_{2}{}^{2})cos\varphi -3y_{2}z_{2}sin\varphi \right] \end{array}\right.$$

Recombine the rotation axis of the permanent magnet with axis Z_2_, and the rotation transformation of O-X_2_Y_2_Z_2_ should also be considered in the rotation process.

Make the coordinate system O-X_2_Y_2_Z_2_ rotate the angle *β* around Z_2_ to obtain the new coordinate system O-X_3_Y_3_Z_3_. The coordinate system transformation can be expressed as:(12)$$\left[\begin{array}{c} x_{3}\\ y_{3}\\ z_{3} \end{array}\right]=R_{z}\cdot \left[\begin{array}{c} x_{2}\\ y_{2}\\ z_{2} \end{array}\right]=\left[\begin{array}{ccc} cos\beta & sin\beta & 0\\ -sin\beta & cos\beta & 0\\ 0 & 0 & 1 \end{array}\right]\cdot \left[\begin{array}{c} x_{2}\\ y_{2}\\ z_{2} \end{array}\right]$$

Therefore, coordinate system O-X_3_Y_3_Z_3_ can be expressed by coordinate base X_2_Y_2_Z_2_ as:(13)$$\left\{\begin{array}{c} x_{3}=x_{2}cos\beta +y_{2}sin\beta \\ y_{3}=-x_{2}sin\beta +y_{2}cos\beta \\ z_{3}=z_{2} \end{array}\right.$$

Substituting it into Equation (11), the component of magnetic field intensity in the coordinate system O-X_3_Y_3_Z_3_ can be expressed by the coordinate basis X_2_Y_2_Z_2_ as:(14)$$\left\{\begin{array}{c} B_{x_{3}}=3B^{\prime }\cdot \left[(x_{2}z_{2}cos\beta +y_{2}z_{2}sin\beta )cos\varphi -(x_{2}cos\beta +y_{2}sin\beta )(y_{2}cos\beta -x_{2}sin\beta )sin\varphi \right]\\ B_{y_{3}}=B^{\prime }\cdot \left[6(y_{2}z_{2}cos\beta -x_{2}z_{2}sin\beta )cos\varphi +{\left(x_{2}cos\beta +y_{2}sin\beta \right)}^{2}sin\varphi -2{\left(y_{2}cos\beta -x_{2}sin\beta \right)}^{2}sin\varphi +z^{2}\right]\\ B_{z_{3}}=B^{\prime }\cdot \left[(2z_{2}{}^{2}-y_{2}{}^{2}-x_{2}{}^{2})cos\varphi -3y_{2}z_{2}sin\varphi \right] \end{array}\right.$$

According to the projection relationship between coordinate systems, the component of magnetic field intensity in coordinate system O-X_2_Y_2_Z_2_ can be expressed as:(15)$$\left\{\begin{array}{c} B_{x_{2}}=B_{x_{3}}cos\beta -B_{y_{3}}sin\beta \\ B_{y_{2}}=B_{x_{3}}sin\beta +B_{y_{3}}cos\beta \\ B_{z_{2}}=B_{z_{3}} \end{array}\right.$$

By replacing the above equation back into Equation (14), the magnetic field intensity component of the magnetic field measured in the coordinate system O-X_3_Y_3_Z_3_ can be expressed in the coordinate system O-X_2_Y_2_Z_2_ as:(16)$$\left\{\begin{array}{c} B_{x_{2}}=B^{\prime }\cdot \left[(2x_{2}{}^{2}-y_{2}{}^{2}-z_{2}{}^{2})cos\beta sin\varphi +3x_{2}y_{2}sin\beta sin\varphi +(6-3cos^{2}\beta )x_{2}z_{2}cos\varphi -3y_{2}z_{2}sin\beta cos\beta cos\varphi \right]\\ B_{y_{2}}=B^{\prime }\cdot \left[3x_{2}y_{2}cos\beta sin\varphi +(2y_{2}{}^{2}-x_{2}{}^{2}-z_{2}{}^{2})sin\beta sin\varphi +3y_{2}z_{2}(1+cos^{2}\beta )cos\varphi -3x_{2}z_{2}sin\beta cos\beta cos\varphi \right]\\ B_{z_{2}}=B^{\prime }\cdot \left[(3x_{2}z_{2}cos\beta +3y_{2}z_{2}sin\beta )sin\varphi +(2z_{2}{}^{2}-x_{2}{}^{2}-y_{2}{}^{2})cos\varphi \right] \end{array}\right.$$

If the permanent magnet rotates in a horizontal plane, which means the direction of the magnetic moment is perpendicular to the Z_2_-axis ($\varphi =\frac{\pi }{2}$), then Equation (16) can be expressed as:(17)$$\left\{\begin{array}{c} B_{x_{2}}=B^{\prime }\cdot \left[(2x_{2}{}^{2}-y_{2}{}^{2}-z_{2}{}^{2})cos\beta +3x_{2}y_{2}sin\beta \right]\\ B_{y_{2}}=B^{\prime }\cdot \left[3x_{2}y_{2}cos\beta +(2y_{2}{}^{2}-x_{2}{}^{2}-z_{2}{}^{2})sin\beta \right]\\ B_{z_{2}}=B^{\prime }\cdot \left[3x_{2}z_{2}cos\beta +3y_{2}z_{2}sin\beta \right] \end{array}\right.$$

If the magnetic moment of the permanent magnet is in the X_2_OY_2_ plane and rotates with angular velocity ω around the Z_2_ axis, the magnetic field intensity component can be expressed as:(18)$$\left\{\begin{array}{c} B_{x_{2}}=B^{\prime }\cdot \left[(2x_{2}{}^{2}-y_{2}{}^{2}-z_{2}{}^{2})cos\omega t+3x_{2}y_{2}sin\omega t\right]\\ B_{y_{2}}=B^{\prime }\cdot \left[3x_{2}y_{2}cos\omega t+(2y_{2}{}^{2}-x_{2}{}^{2}-z_{2}{}^{2})sin\omega t\right]\\ B_{z_{2}}=B^{\prime }\cdot \left[3x_{2}z_{2}cos\omega t+3y_{2}z_{2}sin\omega t\right] \end{array}\right.$$

### 2.2. Phase-Based Goniometric Method

The directivity function of the horizontal directional map of the voltage-beacon antenna is:(19)$$F(\theta_{0})=1+msin\theta_{0}(0\;<\;m\;\leq \;1)$$

If an equal amplitude wave signal is fed to the antenna, the expression is:(20)$$e_{T}(t)=E_{T}sin\Omega t$$

The equal-amplitude wave signal is modulated to obtain a heart-shaped field pattern, and then this heart-shaped field pattern is made to rotate at an angular velocity of Ω. So, the working principle of the phase-based goniometric method is shown in Figure 3. The electrical signal received by the aircraft in the angular $\theta_{0}$ direction can be expressed as:(21)$$e_{t}(t,\theta_{0})=E_{T}(1-msin(\Omega t-\theta_{0}))sin\omega t$$

As shown in Figure 3, the heart-shaped field pattern rotates uniformly with a period T and the target can receive a reference signal when its maximum value points in the positive direction of the Y-axis. If the time elapsed from the receipt of the reference signal to the first receipt of the maximum value of the heart-shaped field pattern by the target is t, then the orientation of the target relative to the signal source can be expressed as:(22)$$\theta_{0}=((T-t)/T)\times 2\pi$$

It can be seen from Equation (21) that there is a one-to-one correspondence between the phase and the angle $\theta_{0}$ of the signal in different directions. From the expression of the rotating magnetic field, it can be concluded that a low frequency periodic magnetic signal can be obtained by combining several different magnetic fields in a certain way, which has a format similar to the above equation. Its envelope waveform can be decomposed as:(23)$$e(t,\omega ,\phi )=a+m_{1}sin(\omega_{1}t+\phi_{1})+m_{2}sin(\omega_{2}t+\phi_{2})+\ldots$$

## 3. Structural Design of Rotating Permanent Magnet Beacon

According to the requirements of the phase-shift angle measurement method, the waveform of the total magnetic field generated by the magnetic beacon in polar coordinates should be similar to the heart shape, and the representation of the waveform converted to the cartesian system should be that there is only one maximum value within a period, and the amplitude on both sides of the maximum value should be as small as possible.

From Equations (18) and (23), we know that the angular velocity $\omega_{n}$ in Equation (23) is jointly determined by the rotational speed of each permanent magnet in the array; the initial phase $\phi_{n}$, constant a and amplitude factor $m_{n}$ are jointly determined by the distance from each permanent magnet in the array to the measurement point and the initial angle of rotation. However, Equation (18) can only represent the magnitude of the magnetic field strength, but not the vector direction of the magnetic field. Therefore, among the above determining factors, the distance from the measurement point to the permanent magnet mainly affects the amplitude of the synthesized signal, while the influence of the rotation speed ω and the initial angle φ on the timing of the magnetic field synthesis at the N and S poles of the permanent magnet is the most important factor of waveform synthesis. Therefore, the optimized design focuses on analyzing the effect of rotation speed and initial angle on the waveform, and to simplify the structural design all models choose symmetric structure.

In this section, according to the current experimental conditions, the structures of the binary array, ternary array and quaternary array are designed first. The mathematical models of the corresponding structures are constructed according to Equation (18), the parameters are adjusted and optimized, and then the optimized structures are simulated and verified by COMSOL physical simulation software.

In advance, the size of the permanent magnet is 13 cm×5 cm×2 cm, and the sintered NdFeB permanent magnet of material grade N_38_ is selected, and its performance parameters are shown in Table 1. Therefore, the residual magnetic field strength of the permanent magnet is set to 1.25T during modeling.

### 3.1. Binary Array Structure

A binary array structure consisting of two permanent magnets is discussed first. As shown in Figure 4, two magnets rotate at angular velocities and $\omega_{2}$ in the XOY plane at (−d_1_, 0) and (d_2_, 0), respectively, and the measurement point P(2,−8,3) is set outside the structure. The angular velocity of rotation, initial rotation angle, and permanent magnet spacing are adjusted individually for simulation. 

According to the experimental conditions and the practical requirements of data processing, first make the rotation speed of the two permanent magnets $\omega_{1}=\omega_{2}=8\pi \;rad/s$, then let the initial angle $\varphi_{1}=\varphi_{2}=0$ and adjust the distance d_1_, d_2_ of the permanent magnet from the origin so that the effect of the spacing d on the rotating magnetic signal synthesis is observed, and the simulation results are shown in Figure 5.

As can be seen from Figure 5, with the increase of the spacing, the magnetic field intensity measured continuously at the fixed-point decreases, and the signal characteristics continue to weaken, but the signal phase does not move and the waveform largely remains the same, so the change of the spacing has no effect on the phase information of the signal and has a greater impact on the propagation distance of the signal. In order to ensure the consistency of the simulation data and the physical measurement data, considering the size of the experimental field, the simulation is fixed with the permanent magnet spacing $d_{1}=d_{2}=4\;m$, and the rotation speed of only one of the permanent magnets is changed to obtain the relationship between the magnetic field strength and the rotation speed at the measurement point, as shown in Figure 6.

It shows that when $\omega_{2}=2\omega_{1}=8\pi \;rad/s$, the signal has fewer extreme points at the same time while the intensity is guaranteed, which is more in line with the signal characteristics required for phase-based goniometry. Then, fixing the permanent magnet pitch and rotation speed and only changing the initial angle $\varphi_{2}$ of the permanent magnet rotation for simulation, the magnetic field strength curve is shown in Figure 7.

By analyzing the simulation results in Figure 7, we can know that with the increase of the initial angle of rotation $\varphi_{2}$, the magnetic field intensity tends to increase and then weaken. When $\varphi_{1}=0,\varphi_{2}=\frac{4\pi }{5}\;rad$, the peak value of the signal is the largest, and the difference between two adjacent great values is more obvious and characteristic. In summary, it can be concluded that the performance of the synthesized signal is better when the two permanent magnets are symmetrically distributed at an interval of 8m, and the rotation speed satisfies $\omega_{2}=2\omega_{1}=8\pi \;rad/s$ and the initial angle of rotation satisfies $\varphi_{1}=0,\varphi_{2}=\frac{4\pi }{5}\;rad$.

The optimized binary array structure parameters are input into the COMSOL software for modelling, and the magnetic field intensity is also collected at P(2,−8,3). The comparison of the magnetic field intensity between the mathematical model and COMSOL model is shown in Figure 8.

It can be seen in the figure that the signal waveforms generated by beacons constructed by the two models are basically the same, among which the main error exists at the minimum value, and the maximum error value is 1.4 nT, and the phase information corresponding to the maximum value does not change, so it can be considered that the design idea of the binary array structure is feasible.

### 3.2. Ternary Array Structure

A binary array structure with good performance is obtained through simulation. On this basis, in order to make the signal intensity greater, the waveform more characteristic, and the phase information more easily extracted, a permanent magnet is added to form a ternary array structure, as shown in Figure 9. From the simulation data of the binary array structure, we can know that the spacing of the permanent magnets only has a large impact on the intensity of the signal and a small impact on the phase information, so on the basis of the binary array structure, the rotation speed and the initial angle of rotation of the permanent magnets are adjusted respectively. Three permanent magnets rotate in the XOY plane at angular velocities $\omega_{1}$,$\omega_{2}$,$\omega_{3}$ at (−d_1_, 0), (d_2_, 0), and (0, d_3_), respectively, and measurement points P(2,−8,3) are set outside the structure, where $d_{3}=\sqrt{3}d_{1}=\sqrt{3}d_{2}=4\sqrt{3}\;m$.

In the first place, the rotation speed of the permanent magnet is optimized for simulation. On the basis of binary array structure, first fix the angular velocities $\omega_{2}=2\omega_{1}=8\pi \;rad/s$, then adjust $\omega_{3}$ to get the relationship between magnetic field strength and rotation speed out of the measurement point, as shown in Figure 10.

According to the simulation data, it can be seen that when $\omega_{2}=2\omega_{2}=2\omega_{1}=8\pi \;rad/s$, the signal has a greater difference in amplitude between adjacent extreme points while ensuring the intensity, and the special rectification is stronger, which is more consistent with the requirements of signal characteristics. Then, fix the pitch and rotation speed of the permanent magnet, adjust the rotation initial angle $\varphi_{3}$ and $\varphi_{2}$ in turn, and get the magnetic field intensity change curve, as shown in Figure 11.

From the simulation results, we get that when $\varphi_{1}=0,\varphi_{2}=\frac{4\pi }{5}\;rad,\varphi_{3}=\pi \;rad$, the signal peak is maximum and the difference between two adjacent great values is more obvious and characteristic. To sum up, the signal performance is better when the three permanent magnets are distributed in a positive triangle at an interval of 8m, the rotation speed satisfies $\omega_{2}=2\omega_{1}=2\omega_{3}=8\pi \;rad/s$, and the initial rotation angle satisfies $\varphi_{1}=0,\varphi_{2}=\frac{4\pi }{5}\;rad,\varphi_{3}=\pi \;rad$.

The optimized ternary array structure parameters are input into the COMSOL software for modelling, and the magnetic field intensity is also collected at P(2,−8,3). The comparison of the magnetic field intensity between the mathematical model and COMSOL model is shown in Figure 12.

It can be seen from the figure that the signal waveforms generated by beacons constructed by the two models are basically the same, among which the main error still exists at the minimum value, the maximum error value is 1.44 nT, and the phase information corresponding to the maximum value is slightly delayed. However, the slope around the maximum value of the signal obtained by the COMSOL software simulation is greater, which is more conducive to the identification of the signal, so it still can be considered that the design idea of the binary array structure is feasible.

### 3.3. Quaternary Array Structure

Based on the ternary array structure, we continue to add a permanent magnet and adjust the rotation speed and the initial angle of rotation of the permanent magnet, respectively. As shown in Figure 13, four permanent magnets are located in the XOY plane at (−d_1_, −d_1_), (d_2_, −d_2_), (−d_3_, d_3_), (d_4_, d_4_), rotating at angular velocities $\omega_{1}$, $\omega_{2}$, $\omega_{3}$, $\omega_{4}$, and with measurement points P(2,−8,3) outside the structure, where $d_{1}=d_{2}=d_{3}=d_{4}=4\;m$.

Primarily, the rotation speed of the permanent magnet is optimized for simulation. According to the design experience, the quadratic array structure is designed by first fixing $\omega_{2}=2\omega_{1}=2\omega_{3}=8\pi \;rad/s$ and then adjusting $\omega_{4}$ to obtain the relationship between the magnetic field strength and rotation speed at the measurement point, as shown in Figure 14.

Comparing the ternary array structure and the quadratic array structure, it can be seen that after adding a permanent magnet, the signal change pattern corresponding to different rotational speeds is basically the same, and when $\omega_{2}=\omega_{4}=2\omega_{1}=2\omega_{3}=8\pi \;rad/s$, the intensity of the signal is relatively larger and the characteristics are more obvious. Via fixed structure spacing, rotation speed, and sequentially adjusting the rotation initial angle $\varphi_{4}$, $\varphi_{3}$, we get the magnetic field intensity change curve as shown in Figure 15.

As can be seen from the figure, by adjusting the initial angle of rotation, the peak of the signal is the largest when $\varphi_{1}=0,\varphi_{2}=\frac{4\pi }{5}\;rad,\varphi_{3}=\frac{6}{5}\pi \;rad,\varphi_{4}=\pi \;rad$, and the difference between two adjacent great values is more obvious and characteristic. On the whole, it can be concluded that the performance of the synthesized signal is better when the four permanent magnets are distributed in a square with 4m interval between two, the rotation speed satisfies $\omega_{2}=\omega_{4}=2\omega_{1}=2\omega_{3}=8\pi \;rad/s$, and the rotation initial angle satisfies $\varphi_{1}=0,\varphi_{2}=\frac{4\pi }{5}rad,\varphi_{3}=\frac{6}{5}\pi \;rad,$$\varphi_{4}=\pi \;rad$.

The optimized quadratic array structure parameters are input into the COMSOL software for modelling, and the magnetic field intensity is also collected at P(2,−8,3). The comparison of the magnetic field intensity between the mathematical model and COMSOL model is shown in Figure 16.

It can be seen from the figure that the signal waveforms generated by beacons constructed by the two models are basically the same, among which the errors occur at both maximum and minimum values, and the maximum error value is 15 nT, and the phase at the maximum point is slightly ahead, so it can be considered that with the increase of the number of permanent magnets, the error between the mathematical model and the simulation model becomes larger, and the practicability of the optimized structure is controversial.

According to Figure 8, Figure 12, and Figure 16, it can be seen that the beacon structural parameters optimized by the mathematical model are applied to the simulation model. Although the law of signal waveform is basically the same, with the increase of the number of permanent magnets, the error at the minimum and maximum amplitude becomes larger and larger. The comprehensive analysis reason is as follows: the mathematical model is based on the magnetic dipole model, but the rectangular permanent magnet model is established in the simulation verification. In fact, the distribution pattern of the magnetic field of a rectangular permanent magnet in the near field is different from that of a magnetic dipole, so the magnetic signal waveform generated by the two models is basically the same, but there will be errors in the calculation at the extreme point. At the same time, by analyzing the magnetic field of ternary array and quadratic array structures, it can be seen that increasing the number of permanent magnets can significantly improve the strength of a magnetic signal, but the magnetic field intensity on both sides of the maximum point will continue to increase, thus weakening the characteristics of the signal. To sum up, for the validity of the subsequent experimental results, the ternary array structure is selected as the experimental verification object.

## 4. Analysis of the Effect of Rotating Permanent Magnetic Beacons on Different near Fields

The magnetic field of permanent magnets decays rapidly with the increase of distance, and the simulation calculation can conclude that increasing the number of permanent magnets to form an array can effectively increase the magnetic field strength. In practice, the rotating magnetic beacon may be used in various environments, and the propagation medium of the magnetic signal is not only limited to air, so the ternary array structure is chosen here, and the propagation medium of the magnetic signal is replaced by air, soil, and seawater, respectively, and the field strength change of the structure in different media is obtained by using simulation software. By reviewing the data, it is known that the relative permeability of air is 1, that of soil is slightly greater than 1, and that of seawater is slightly less than 1. The field strength variation curves obtained by setting different permeabilities are shown in Figure 17.

When changing the propagation medium, the magnetic signal intensity changes with distance in the same pattern, but due to the difference in relative permeability, the magnetic signal attenuation increases in seawater and decreases in soil, which is greatly related to the composition of the medium, but has little effect on the signal propagation trend, so the magnetic beacon structure proposed in this paper can play a stable effect in common media.

## 5. Experiment and Analysis

Since there are differences between the permanent magnets simulated by the software and those obtained by actual machining, further experiments are needed to verify the performance of the array structure obtained by simulation optimization.

### 5.1. Beacon Array Layout and Signal Testing

The dimensions of the three permanent magnets in the magnetic beacon array are the same, *L* × *W* × *H* = 13 cm × 5 cm × 2 cm, the remanent magnetization of the permanent magnets is 1.25 T and the demagnetization coefficient is 0.14. The layout of the array is shown in Figure 18. The three permanent magnets rotate along the Z-axis clockwise with angular velocities of $\omega_{1}$, $\omega_{2}$ and $\omega_{3}$, respectively. The spacing of all three permanent magnets is d, and the initial angles of rotation are $\varphi_{1}$, $\varphi_{2}$, and $\varphi_{3}$. The measurement point is set at a point P outside the array, and the synthetic magnetic field generated by the array is measured after steps such as noise reduction and offsetting the ambient magnetic field.

### 5.2. Experiment and Result Analysis

A measurement system platform is built in the laboratory, which consists of the following parts: a magnetic beacon, non-magnetic turntable, three-axis magnetometer sensor, AC/DC power supply, signal acquisition, and data processing software, etc., as shown in Figure 19.

The permanent magnets used in the experiment are the same as those in the simulation design, and the individual permanent magnets are shown in Figure 20. According to the design scheme, the coordinates of the turntable in the measurement array are (−4,0,0), (4,0,0), (0,$4\sqrt{3}$,0), where the midpoint of turntable 1 and turntable 2 is the coordinate origin, and the rotation speed of the turntable is set to $\omega_{1}=4\pi \;rad/s$, $\omega_{2}=8\pi \;rad/s$, $\omega_{3}=4\pi \;rad/s$, respectively, and the sensor is placed at coordinate P(2,−8,3), as shown in Figure 21.

After adjusting the rotation parameters, the power supply is started, the three rotary tables are rotated simultaneously, the measurement software is started, and the magnetic field information is collected using a three-axis flux meter with a sampling rate of 200 Hz, the data for which mainly includes: acquisition time, total magnetic field *B*, magnetic field component *B_x_*, magnetic field component *B_y_*, and magnetic field component *B_z_*. The collected data are imported into Origin software for waveform plotting, and the experimental process is shown in Figure 22.

As shown in Figure 23, the magnetic signal is collected with a three-axis flux meter, and the waveform is stabilized and selected for plotting within 1s time. An error analysis of experimental and theoretical values is shown in Figure 24.

As can be seen from Figure 22, at the first maximum point, the experimental value is smaller than the theoretical value; at the second maximum point, the experimental value is larger than the theoretical value; and at the other extreme points, the experimental value is smaller than the theoretical value, and the maximum error is 2.71 nT. After arranging the experimental environment and instruments, the reasons for errors are summarized as follows: (1) By measuring the magnetic field intensity of each permanent magnet, it is found that the three permanent magnets are not completely consistent, and the magnetic value of the permanent magnet at position 3 is smaller than the theoretical value, while the magnetic value of the permanent magnet at position 1 and 2 is larger than the theoretical value. In the process of rotation, the speed of the permanent magnet at position 2 is twice that of the permanent magnet at position 1 and 3. Therefore, in the process of signal superposition, the signal is too large when it should be cancelled and too small when it should be enhanced; (2) the switches of the three turntables are not controlled at the same time, so there will be time deviation, which will lead to an error at the moment when the extreme point appears; (3) the time from start-up to stability of the turntable is different, so in the process of rotation, the initial rotation angle is not completely consistent with the theoretical value, which leads to the deviation of the magnetic field intensity at the corresponding moment of the extreme point; (4) the laboratory is not a non-magnetic environment. Because the indoor power supply mode is an alternating current, the interference magnetic field with variable intensity and frequency will also be generated, thus affecting the experimental results.

Although the magnetic signal measured via experimentation has some errors compared with the theoretical value, the error value is within the permissible range, and the overall change law of the signal is consistent with the change law of the theoretical value, so it can be considered that the magnetic beacon designed in this paper has certain practicability.

In order to further verify the feasibility of the method, this paper will use the physical simulation and theoretical simulation method to experiment with the direction finding function. The following improvements are made to solve the problems that emerged from the actual testing of the fiducial model: (1) Ensure that the permanent magnets are consistent with the theoretical model before the experiment; (2) adjust the control system of the turntable to ensure the synchronization of rotation as much as possible; (3) add a random resonance detection method to the data processing to maximize the effect of the ambient magnetic field while also improving the SNR of the output signal.

Next, the orientation experiment is performed. First, set up the magnetic beacon; as shown in Figure 25, the center of the array O is the center of the circle; then, set five test points in the radius of 8 m on the circumference: P_1_($4\sqrt{3}$,−4,3), P_2_(4,−$4\sqrt{3}$,3), P_3_(0,8,3), P_4_(−4,−$4\sqrt{3}$,3), P_5_(−$4\sqrt{3}$,−4,3). The signal receiving and processing unit will be placed at these five test points in turn, after the turntable start and smooth rotation to start collecting magnetic signals.

When the measurement point is at P_1_ ($4\sqrt{3}$,−4,3), the maximum value of the magnetic signal needs to rotate through 120° from the positive half-axis of the Y-axis to the measurement point, corresponding to a time of 0.167 s, and the actual measurement results are shown in Figure 26.

Based on the measurement results, we know that the time difference from the reference to the first maximum point of the signal is 0.164 s, corresponding to an angle of 118.08°.

When the measurement point is at P_2_(4,−$4\sqrt{3}$,3), the maximum value of the magnetic signal needs to rotate through 150° from the positive half-axis of the Y-axis to the measurement point, corresponding to a time of 0.208 s, and the actual measurement results are shown in Figure 27.

Based on the measurement results, we know that the time difference from the reference to the first maximum point of the signal is 0.201 s, corresponding to an angle of 147.6°.

When the measurement point is at P_3_(0,8,3), the maximum value of the magnetic signal needs to rotate through 180° from the positive half-axis of the Y-axis to the measurement point, corresponding to a time of 0.25 s, and the actual measurement results are shown in Figure 28.

Based on the measurement results, we know that the time difference from the reference to the first maximum point of the signal is 0.247 s, corresponding to an angle of 177.84°.

When the measurement point is at P_4_(−4,−$4\sqrt{3}$,3), the maximum value of the magnetic signal needs to rotate through 210° from the positive half-axis of the Y-axis to the measurement point, corresponding to a time of 0.291 s, and the actual measurement results are shown in Figure 29.

Based on the measurement results, we know that the time difference from the reference to the first maximum point of the signal is 0.289 s, corresponding to an angle of 208.08°.

When the measurement point is at P_5_(−$4\sqrt{3}$,−4,3), the maximum value of the magnetic signal needs to rotate through 240° from the positive half-axis of the Y-axis to the measurement point, corresponding to a time of 0.334 s, and the actual measurement results are shown in Figure 30.

Based on the measurement results, we know that the time difference from the reference to the first maximum point of the signal is 0.33 s, corresponding to an angle of 237.6°.

The orientation measurement error at the five measurement points can be obtained according to Figure 26, Figure 27, Figure 28, Figure 29 and Figure 30, as shown in Figure 31.

From the figure, we can see that the measurement error is mainly distributed around 2°, and the noisy signal will have some distortion after random filtering detection and smoothing, which leads to deviation in the measurement of the maximum value, thus affecting the measurement of the orientation. When the distance is close, the deviation of 2° is not obvious, but as the distance increases, the angular error will become more and more obvious, so this directional measurement system needs to continue to improve the problem of improving the accuracy, such as the optimization of the beacon structure, the improvement of the signal detection algorithm, and the improvement of the data acquisition software and other aspects.

In general, the experimental system can complete the measurement of target orientation after the initial optimization, which verifies the feasibility of the proposed method to a certain extent.

## 6. Conclusions

To address the problem of difficulty in improving the positioning accuracy in the artificial magnetic beacon positioning based on amplitude measurement, (1) establish the rotating permanent magnet model, derive the expression of magnetic field intensity at any point outside the rotating permanent magnet, and analyze the spatial characteristics of the magnetic field; (2) use the MATLAB software to simulate and analyze the binary array, ternary array and quadratic array structures on the basis of the mathematical model, and find that a certain number of permanent magnets are placed according to the design, and the rotation parameters of each permanent magnet are adjusted to obtain a signal with strong characteristics to meet the phase orientation, but with the increase of the number of permanent magnets, the characteristics of the signal first increase and then decrease. The signal generated by the ternary array structure has stronger characteristics while satisfying the requirements. The COMSOL physical simulation software is used to verify whether the optimized structure obtained via mathematical model is available; (3) three common environments of air, seawater and soil are simulated to verify that the obtained structure can function stably in them; (4) an experimental system is established, and the detection of magnetic beacon signals and the feasibility of using magnetic beacon signals for direction finding are carried out successively. The results indicated that the optimally designed beacons could be applied to the direction finding method and achieve better results. Measuring the phase information is different from measuring the amplitude, which does not require high amplitude accuracy at each moment, providing a good idea to further improve the positioning accuracy, so that a larger size of the permanent magnet can be made to further analyze the working performance at longer distances when the conditions allow.

## Figures and Tables

**Figure 1 sensors-22-08304-f001:**
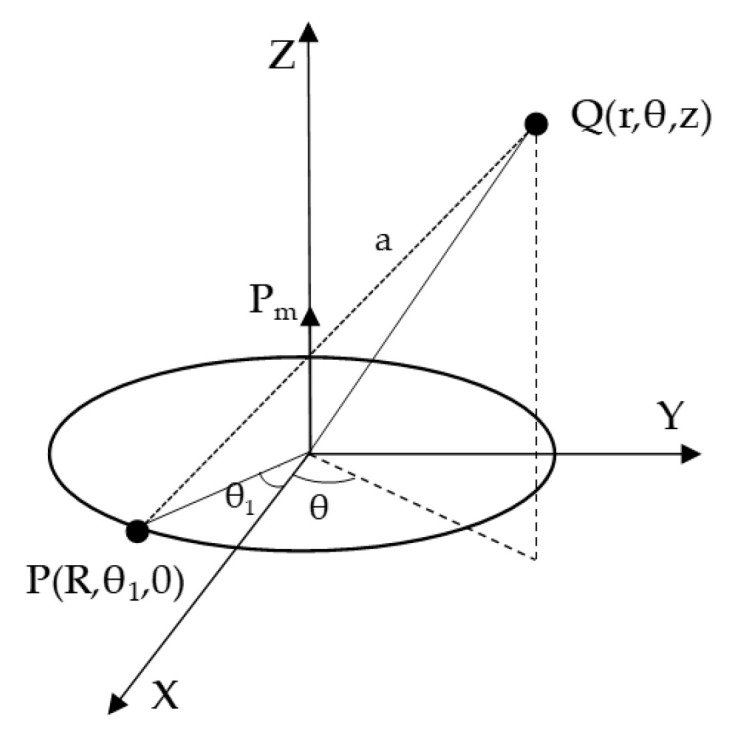
A single circular micro electric current.

**Figure 2 sensors-22-08304-f002:**
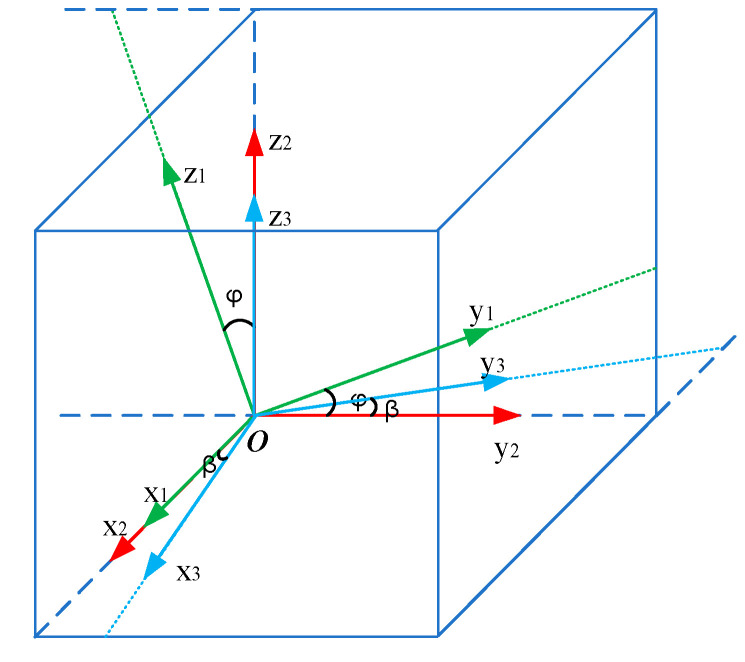
Coordinate transformation.

**Figure 3 sensors-22-08304-f003:**
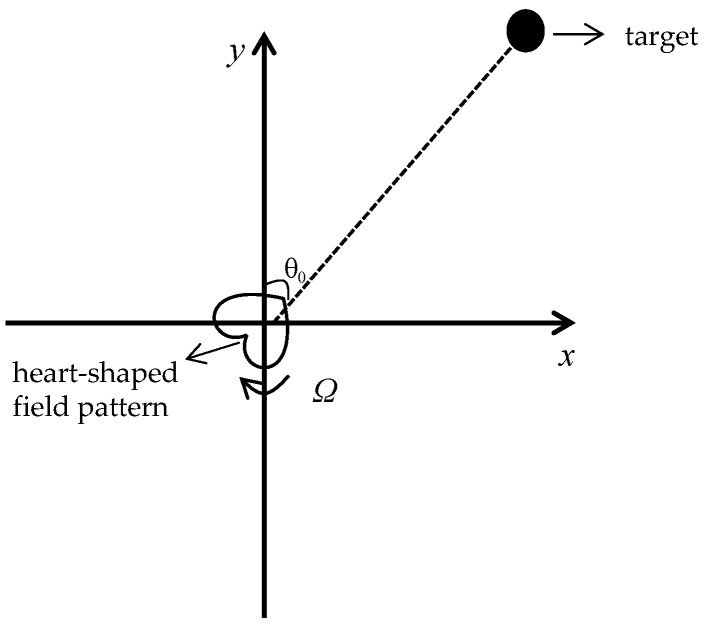
A working principle of the phased-based goniometric method.

**Figure 4 sensors-22-08304-f004:**
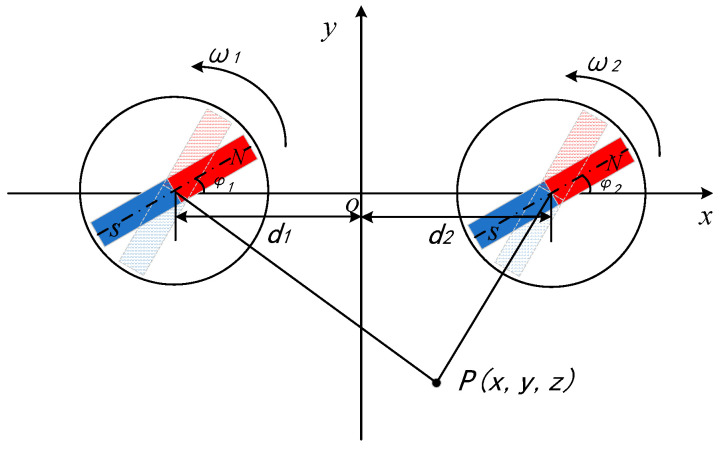
The binary array structure.

**Figure 5 sensors-22-08304-f005:**
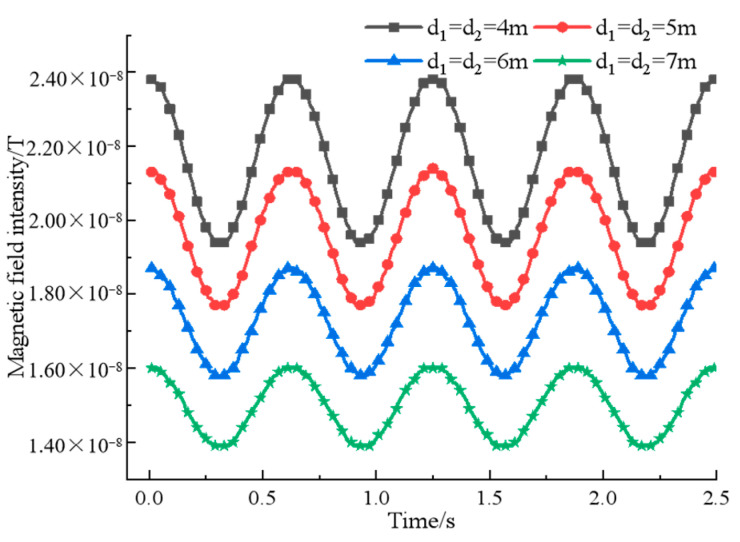
A comparison of the binary array’s magnetic field intensity at different spacing.

**Figure 6 sensors-22-08304-f006:**
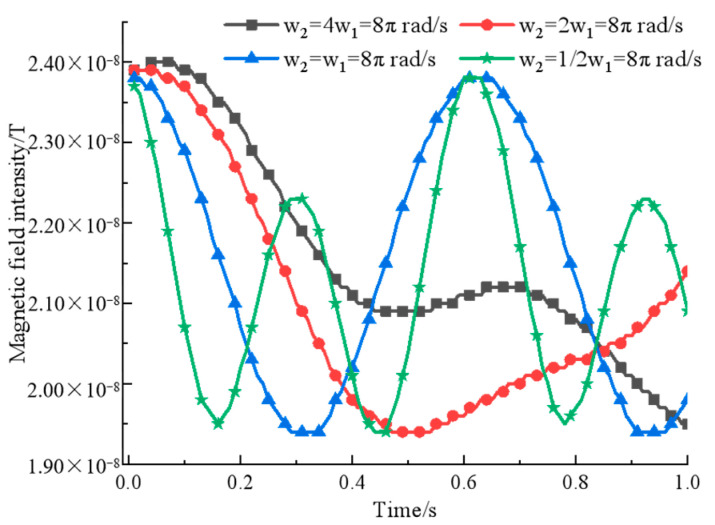
A comparison of binary array’s magnetic field intensity at different rotational speeds.

**Figure 7 sensors-22-08304-f007:**
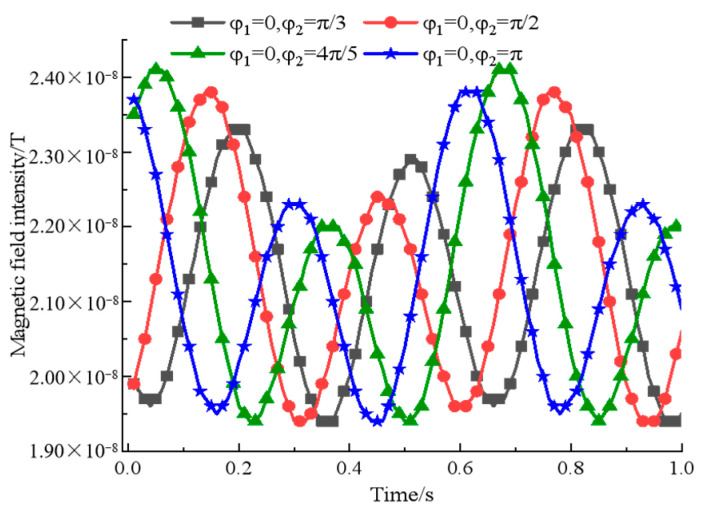
A comparison of binary array’s magnetic field intensity at different initial angles of rotation.

**Figure 8 sensors-22-08304-f008:**
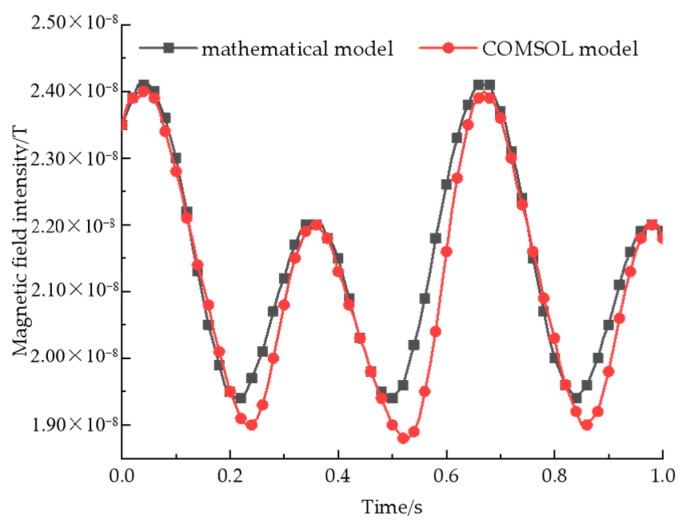
A comparison of magnetic field intensity of a binary array between the mathematical model and the COMSOL model.

**Figure 9 sensors-22-08304-f009:**
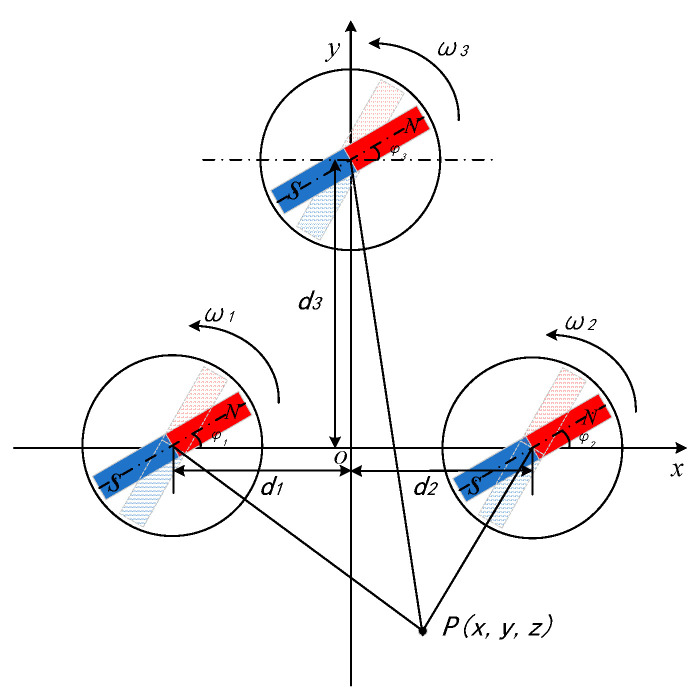
The ternary structure.

**Figure 10 sensors-22-08304-f010:**
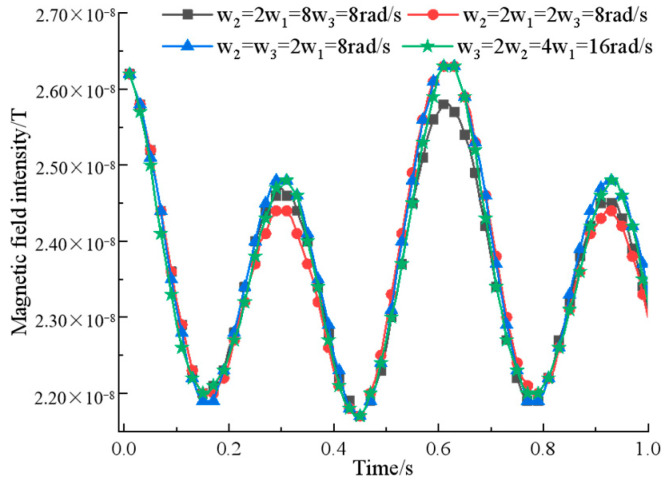
A comparison of ternary array’s magnetic field intensity at different rotational speeds.

**Figure 11 sensors-22-08304-f011:**
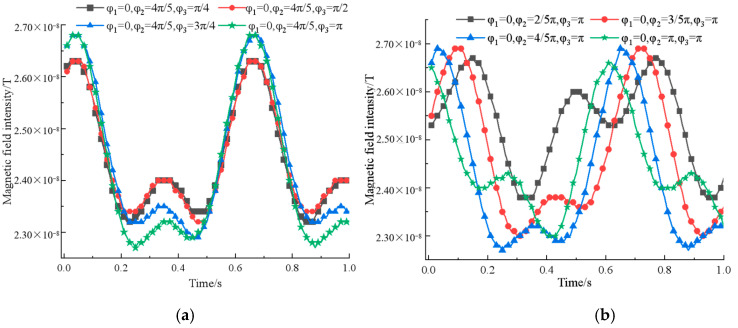
A comparison of ternary array’s magnetic field intensity at different initial angle of rotation: (**a**) Changing the initial rotation angle $\varphi_{3}$; (**b**) changing the initial rotation angle $\varphi_{2}$.

**Figure 12 sensors-22-08304-f012:**
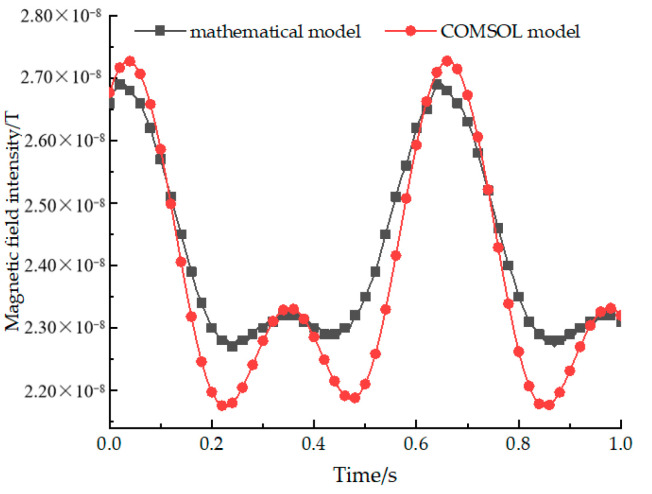
A comparison of the magnetic field intensity of a ternary array between the mathematical and COMSOL models.

**Figure 13 sensors-22-08304-f013:**
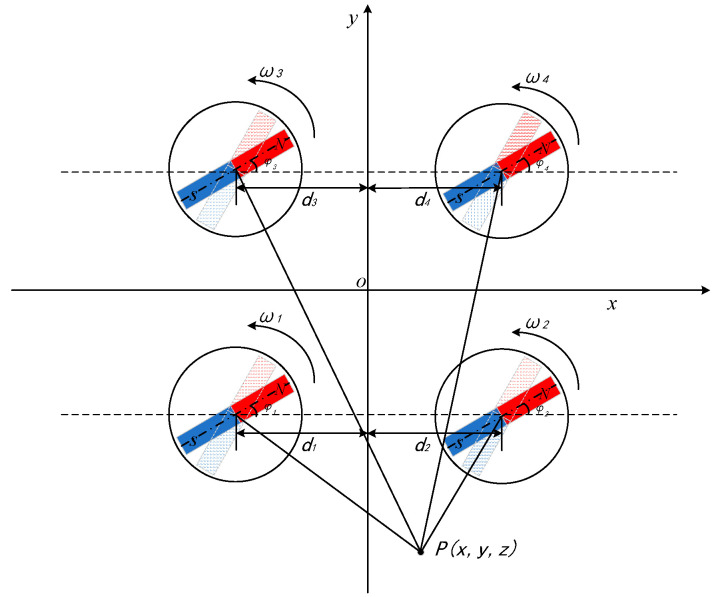
The quaternary array structure.

**Figure 14 sensors-22-08304-f014:**
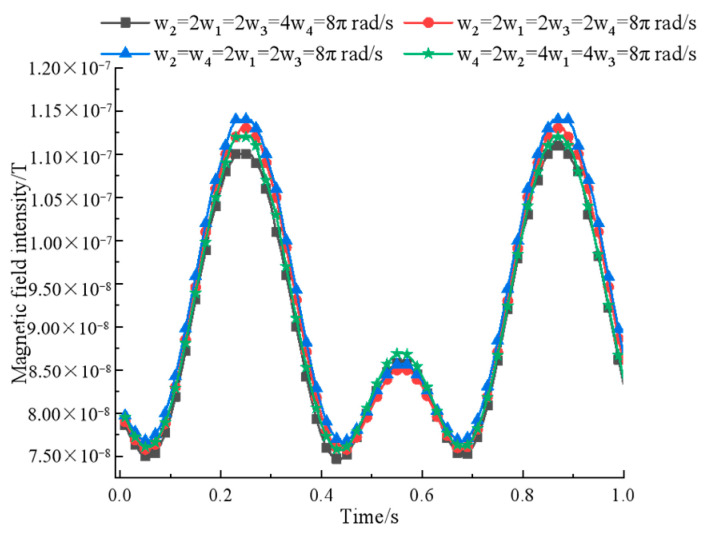
A comparison of quaternary array’s magnetic field intensity at different rotational speeds.

**Figure 15 sensors-22-08304-f015:**
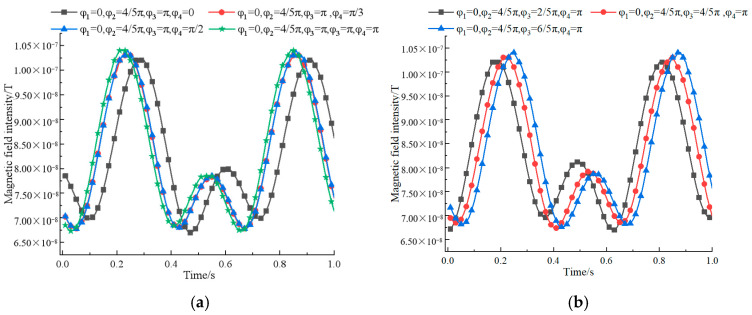
A comparison of quaternary array’s magnetic field intensity at different initial angles of rotation: (**a**) Changing the initial rotation angle $\varphi_{4}$; (**b**) changing the initial rotation angle $\varphi_{3}$.

**Figure 16 sensors-22-08304-f016:**
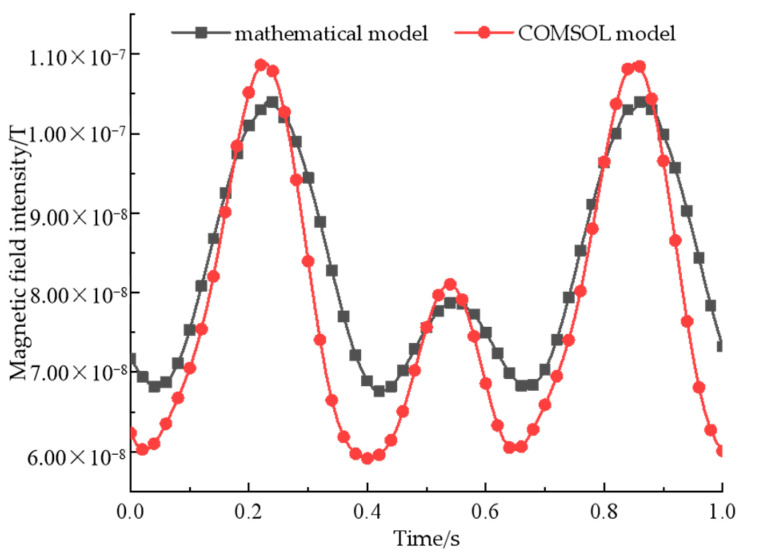
A comparison of magnetic field intensity of the quaternary array between the mathematical and COMSOL models.

**Figure 17 sensors-22-08304-f017:**
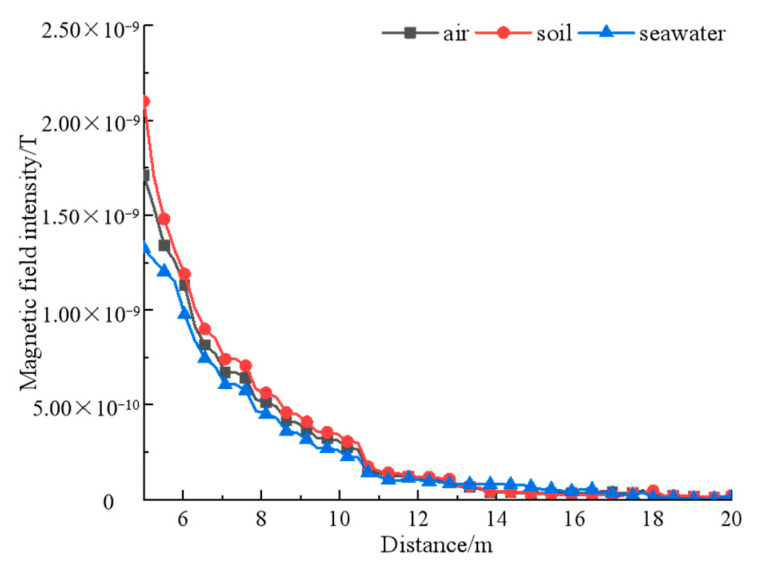
A comparison of magnetic field intensity in different media.

**Figure 18 sensors-22-08304-f018:**
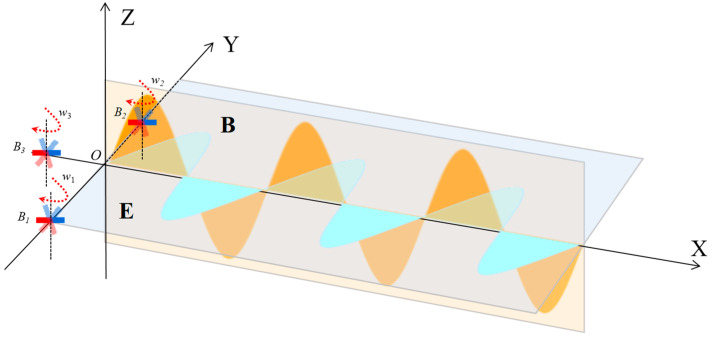
A schematic diagram of beacon operation.

**Figure 19 sensors-22-08304-f019:**
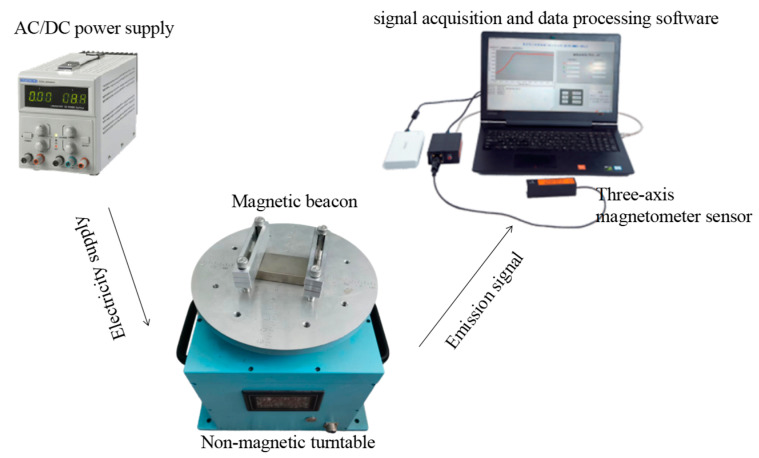
The measurement system platform.

**Figure 20 sensors-22-08304-f020:**
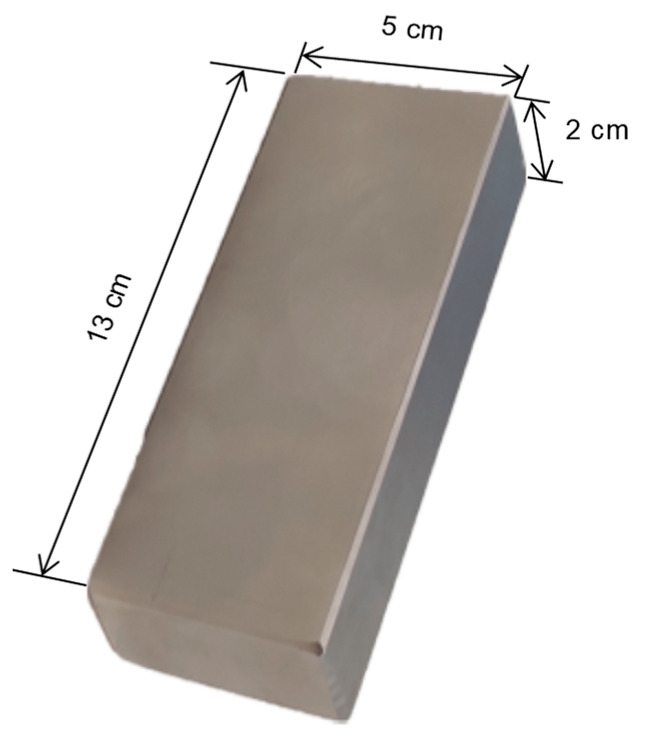
The permanent magnet.

**Figure 21 sensors-22-08304-f021:**
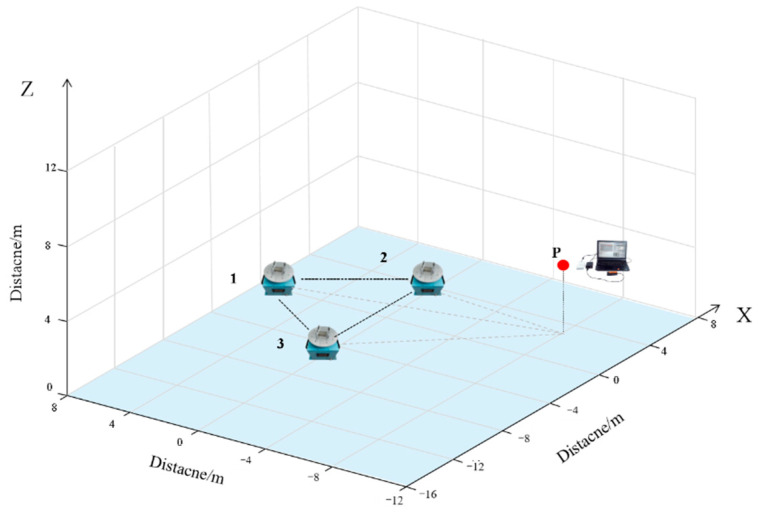
The experimental schematic.

**Figure 22 sensors-22-08304-f022:**
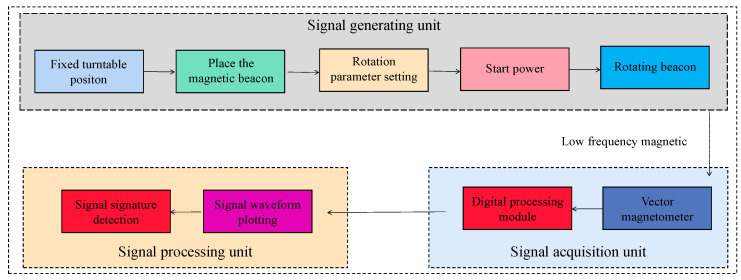
The experimental procedure.

**Figure 23 sensors-22-08304-f023:**
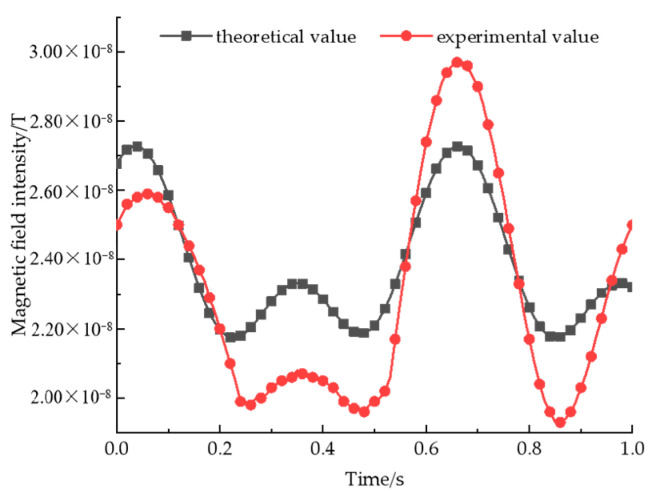
A comparison of magnetic field intensity test results of a ternary array structure.

**Figure 24 sensors-22-08304-f024:**
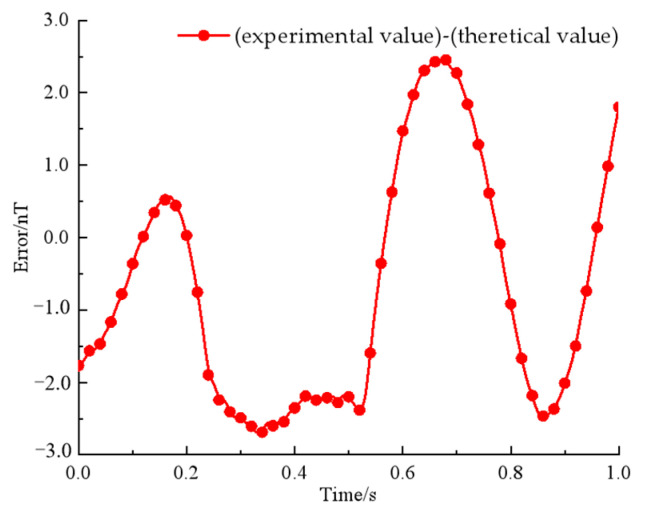
An error analysis.

**Figure 25 sensors-22-08304-f025:**
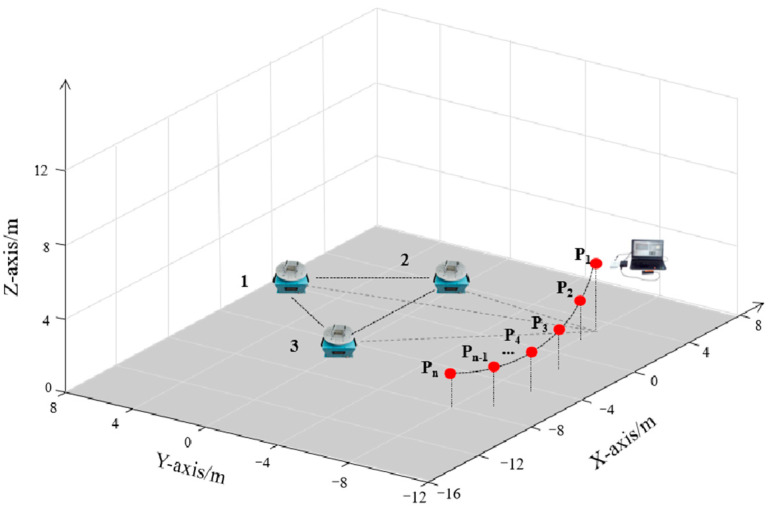
A schematic diagram of the experiment.

**Figure 26 sensors-22-08304-f026:**
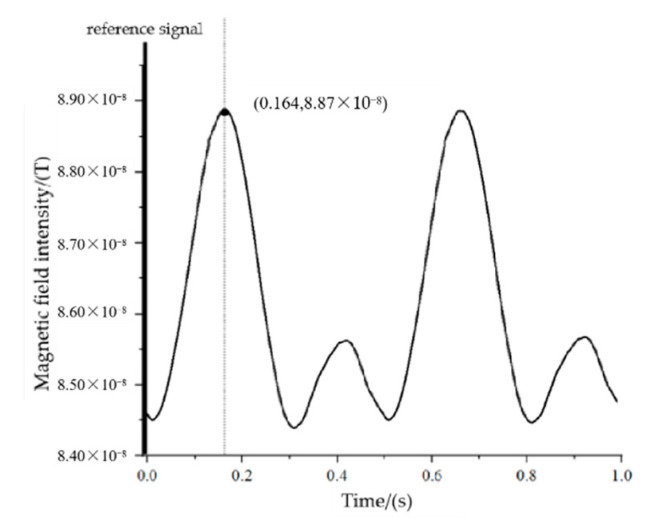
The measured signal at point P_1_.

**Figure 27 sensors-22-08304-f027:**
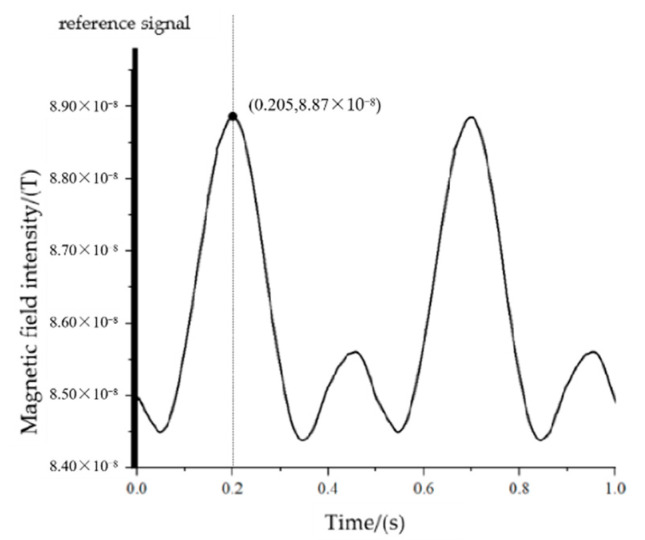
The measured signal at point P_2_.

**Figure 28 sensors-22-08304-f028:**
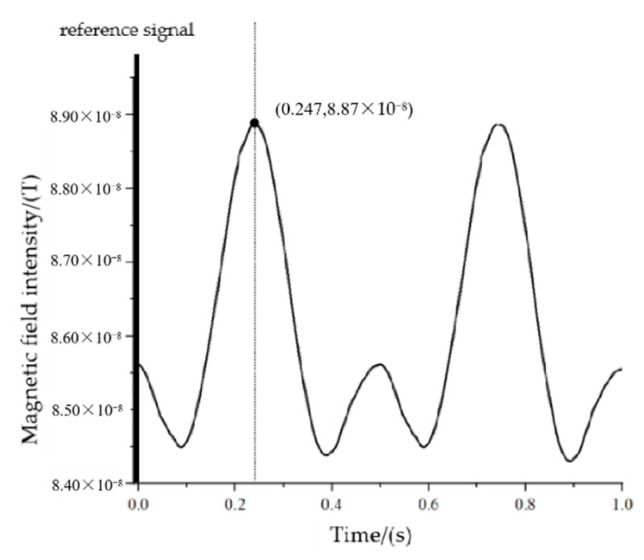
The measured signal at point P_3_.

**Figure 29 sensors-22-08304-f029:**
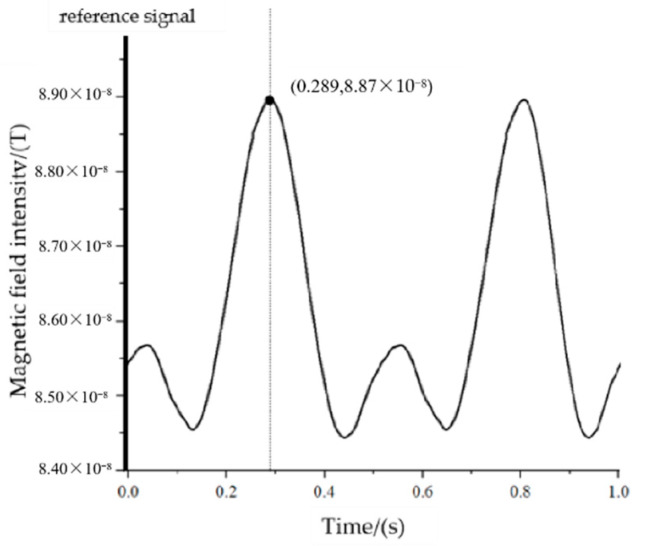
The measured signal at point P_4_.

**Figure 30 sensors-22-08304-f030:**
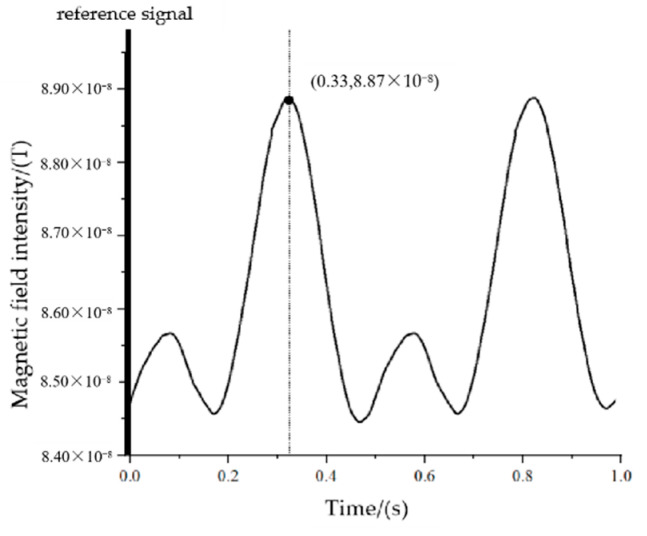
The measured signal at point P_5_.

**Figure 31 sensors-22-08304-f031:**
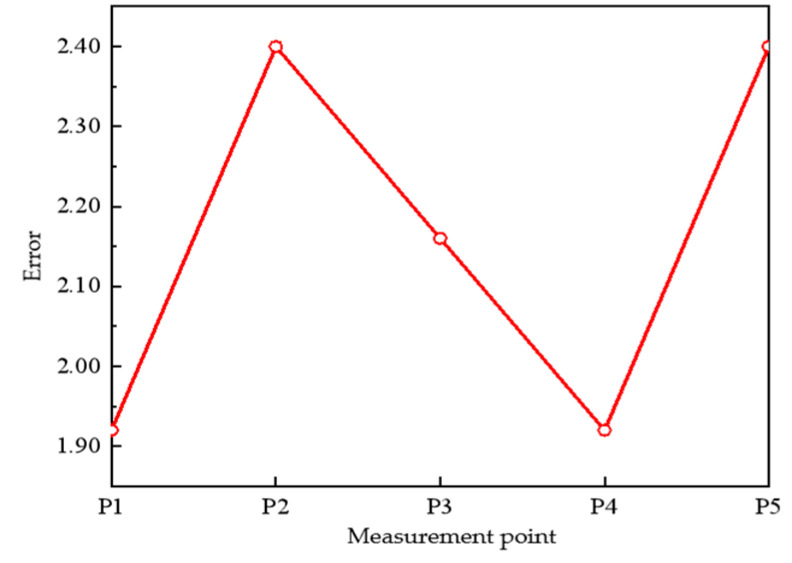
Measuring error.

**Table 1 sensors-22-08304-t001:** The main parameters of the NdFeB.

		Performance Parameter			
Type	Material	*B_r_*/T	*H_cj_*/KA m^−1^	(*BH*)*_m_*/KJ·m^−3^	*T_c_*/°C
Sintered NdFeB	N_38_	1.22–1.26	≥955	287–303	≤80

## Data Availability

Not applicable.

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
