# Peer review of "The Source Structure Design of the Rotating Magnetic Beacon Based on Phase-Shift Direction Finding System"

_sensors, 2022, doi:10.3390/s22218304_

Round 1

Reviewer 1 Report

please find the attached comments.

Author Response

Dear Reviewer,

Reviewer 2 Report

Dear Authors,

I have some comments on your article:

1. At the end of the introduction section there is no information on how the article is organized.

2. Literature should be checked if there are no newer items. Especially from the last 18 months.

3. All indexes in symbols in text and equations should be checked carefully.

4. Please describe the comparison of simulation and measurement results in more detail.

5. Please describe in more detail and indicate in the text the use of the COMSOL software.

6. In the summary of the article, please provide more technical information about the practical possibility of implementing the proposed method.

Author Response

Dear Reviewer,

Round 2

Reviewer 1 Report

Thank the authors for making clear improvements in this submission. I have suggested four aspects of a rigorous theoretical derivation or comprehensive model tests to show that the technique works. The authors still need to do more in the first and fourth aspects.

Major points:   1: In the newly submitted manuscript, the authors have discussed the expected characteristics of the signal generated by the magnetic beacon. And the mathematical formula of the magnetic components created by dipole source have been given before. In my view, then the properties of the magnetic beacon (combination mode, angular velocity, shape, magnetization, etc.) that may affect the signal characteristics should be analyzed before starting designing the array structure of beacon. And then the experimental test should be organized according to the analysis. As I wrote earlier: “only rigorous theoretical derivation or comprehensive experimental simulations can lead to convincing conclusions.” The combination situation of beacon array in this manuscript is limited. The article mentions the desire to generate a signal that “there is only one maximum value within a period and the amplitude on both sides of the maximum value should be as small as possible” (Line 181-182, close to Dirac delta function?). I strongly suggested that the authors analyze what properties of the magnetic beacon affects the signal shape and use it as a guide for the following array design.    2: Passive navigation is a very interesting topic, but the conclusion of the article is vague. Readers may be concerned about whether the new design can be used for navigation, or to what extent can it improve navigation accuracy, which is the purpose and foothold of the research. As another reviewer pointed out: “please provide more technical information about the practical possibility of implementing the proposed method”. I strongly recommend incorporating some theoretical simulations to verify the feasibility of the proposed method if the experimental conditions are limited.

Author Response

Dear Reveiwer,

Please refer to the Word file for comments on modifications.

Kind regards,

Mr. Li

Reviewer 2 Report

Dear Authors,

Thank you very much for introducing changes that have improved the quality of the article. I have no more comments.

Best regards

Author Response

Dear Reviewer,

Thank you very much for your support and recognition of our work.

Kind regards,

Mr. Li

Round 3

Reviewer 1 Report

Thank the authors for making improvements in the new submission. The authors need to refine the experiment in measurement of target orientation. 

As shown in figure 31, the measuring errs of target orientation angle are less than 2 degrees, which is encouraging for the application in negative navigation. The authors have discussed the decay rate of magnetic field intensity in different media in figure 17, and I am interested in whether the magnetic field amplitude information can be used to determine the distance between the target and the source, and to complete the positioning combined with knowledge of the measured target orientation angle.

Author Response

Dear Reviewer,

Please find attached the reply to our suggestion.

Kind regards,

Mr. Li
